# SHARPER-DOSY: Sensitivity enhanced diffusion-ordered NMR spectroscopy

George Peat[1], Patrick J. Boaler[1], Claire L. Dickson [1,2], Guy C. Lloyd-Jones [1] & Dušan Uhrín [1] ✉

Since its discovery in mid-20th century, the sensitivity of Nuclear Magnetic Resonance (NMR) has increased steadily, in part due to the design of new, sophisticated NMR experiments. Here we report on a liquid-state NMR methodology that significantly increases the sensitivity of diffusion coefficient measurements of pure compounds, allowing to estimate their sizes using a much reduced amount of material. In this method, the diffusion coefficients are being measured by analysing narrow and intense singlets, which are invariant to magnetic field inhomogeneities. The singlets are obtained through signal acquisition embedded in short (<0.5 ms) spin-echo intervals separated by non-selective 180° or 90° pulses, suppressing the chemical shift evolution of resonances and their splitting due to $J$ couplings. The achieved 10−100 sensitivity enhancement results in a 100−10000-fold time saving. Using high field cryoprobe NMR spectrometers, this makes it possible to measure a diffusion coefficient of a medium-size organic molecule in a matter of minutes with as little as a few hundred nanograms of material.

NMR spectroscopy has impacted many areas of chemistry, biology and physics through to its ability to separate responses from nuclei of the same type along the chemical shift axis. Combined with another fundamental property of nuclei, scalar couplings, NMR has established itself as the leading technique for liquid-state molecular structure elucidation. However, many applications of NMR are limited by the inherently low sensitivity of the technique.

Splitting of signals due to $J$ couplings reduces the signal-to-noise ratio (SNR) of NMR spectra, hence $^{13}C$ spectra are practically always acquired with $^1H$ decoupling. More recently, so called "pure shift" $^1H$-detected experiments implemented real-time homonuclear decoupling[1–4], aiming to simultaneously reduce spectral overlap and to increase the SNR of liquid-state NMR spectra.

One example, where significant sensitivity gains have been achieved by removing $J$ splittings is SHARPER (Sensitive, Homogeneous And Resolved PEaks in Real time), a technique originally proposed to boost the sensitivity of reaction monitoring[5]. SHARPER removes hetero- and homonuclear splittings of a single selected signal in real time by interrupting data acquisition with 180° refocussing pulses. When non-selective pulses are used, all heteronuclear couplings are removed, while selective 180° pulses also remove homonuclear couplings; in both instances this only requires to pulse on the acquired nucleus. As SHARPER acquisition is embedded within a CPMG pulse sequence[6,7], it eliminates the effects of magnetic field inhomogeneity and generates extremely narrow signals that approach their natural linewidth. Both of these factors contribute to significant sensitivity gains as recently demonstrated on benchtop NMR spectrometers[8].

The original SHARPER experiments acquired signal during spin-echo intervals (referred to as acquisition chunk times, or just chunk times), τ, on the order <0.25/$J$ as required to eliminate $J$ evolution. However, reducing the length of spin-echo intervals below 1.0 ms in combination with non-selective 180° refocusing pulses also removes $J$ evolution[9], the property which underpins the measurement of spin-spin relaxation times by the CPMG pulse sequence[7].

We demonstrate here that a removal of frequency modulation on the chemical shift scale over thousands of Hz is entirely feasible and only requires a further reduction of the spin-echo intervals. The

[1]EaStCHEM School of Chemistry, University of Edinburgh, David Brewster Rd, Edinburgh EH9 3FJ, UK. [2]Present address: Oxford Instruments, Halifax Road, High Wycombe HP12 3SE2, UK. ✉e-mail: dusan.uhrin@ed.ac.uk

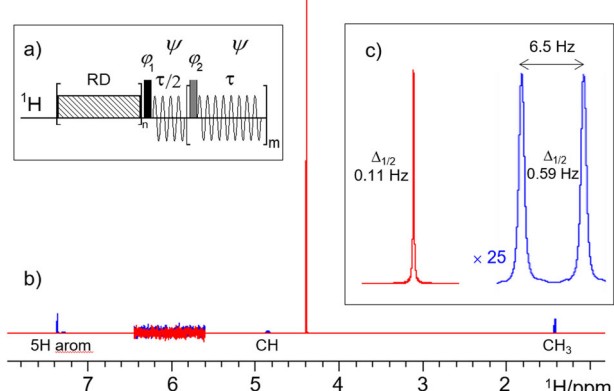

**Fig. 1 | Structures of model compounds used in this study.** 1-phenylethanol, **1**, cyclosporine, **2**, and sodium cholate, **3**.

**Fig. 2 | Collapsing the entire $^1$H NMR spectrum of** 1 **into a sharp singlet. a** Pulse sequence of a non-selective SHARPER experiment (for explanation of symbols and details see Supplementary Note 1.1); **b** overlay of a 400 MHz 1D $^1$H NMR spectrum (blue) of **1** in $D_2O$, and its collapsed spectrum (red) acquired using, $\tau = 200\,\mu s$ and $60\,\mu s$ 180° refocusing pulses. Both real and imaginary SHARPER time domain points were used, no line broadening was applied. The 5.6–6.4 ppm region was scaled up vertically to illustrate that both spectra have identical noise levels; **c** expansions of the SHARPER singlet and the $CH_3$ doublet from the 1D spectrum using identical horizontal scales; the $CH_3$ signal was scaled up 25 times to equal the height of the SHARPER signal.

resulting collapsed signal contains a cumulative response from all selected protons, and serves as an intense molecular signature that can report on a compound concentration, or its molecular property. By applying this concept to the measurement of translational diffusion of molecules on high field NMR spectrometers, we have developed a protocol that boosts the sensitivity of Diffusion Ordered SpectroscopY (DOSY) up to two orders of magnitude, enabling fast and reliable determination of diffusion coefficients at very low sample concentrations. The proposed technique is referred to herein as SHARPER-DOSY. Its performance is demonstrated using three model compounds, 1-phenylethanol, **1**, cyclosporine, **2**, and sodium cholate, **3** (Fig. 1)

## Results

### Collapsing NMR spectra into sharp singlets

Using a benchtop NMR instrument, we have recently presented a SABRE-SHARPER experiment[10], in which the SHARPER acquisition module provided a further boost (up to 17-fold) to a hyperpolarised signal by removing inhomogeneous broadening, refocusing homo- and heteronuclear $J$ couplings and chemical shift differences on the order of 75 Hz. Using the pulse sequence of Fig. 2a, we show here that this approach can be used to collapse signals spanning thousands of Hz.

A 400 MHz $^1$H NMR spectrum of **1** (Fig. 2b) containing signals of nine non-exchangeable protons resonating across 6 ppm, was collapsed using the pulse sequence of Fig. 2a (Supplementary Note 1.1) into a SHARPER singlet shown in red in Fig. 2b. The 1D and SHARPER spectra presented in Fig. 2 were acquired using identical parameters and Fourier transformed without any apodisation. The obtained SHARPER singlet is 25 × taller and 5.4 × narrower than the $CH_3$ doublet of the 1D spectrum (Fig. 2c). The SHARPER time domain points and the resulting spectrum are analysed in detail in Supplementary Note 2.

To rationalise this level of signals enhancement, the efficiency of SHARPER acquisition was investigated. Using the HOD protons of a doped $D_2O$ sample, the SHARPER signal was inspected as a function of the frequency offset, $\Delta\nu$, of the HOD resonance frequency from the carrier frequency and its integral intensity across a 2400 Hz frequency range and chunk times $\tau = 100$, 200 and 400 μs was monitored (Fig. 3a). While practically identical on-resonance, the values decrease gradually to a level of 84, 63 and 3% at 2400 Hz, respectively, with increasing chunk time.

Another representation of the data, which quantifies the height of the signal and takes into account the effective spin-spin relaxation of the SHARPER signal, $T_2^S$, (see Supplementary Note 3) is presented in Fig. 3b. It can be seen that closer to the HOD frequency, the signal intensities are larger for longer chunk times. This is due to slower effective relaxation resulting from less frequent use of refocusing pulses, producing narrower, taller signals and ultimately higher SNRs. When collapsing narrower spectral regions, it is therefore beneficial to use longer chunk times.

Returning to the analysis of the SHARPER spectrum of **1**, its integral represents 87.1% of the integral of the entire reference 1D $^1$H spectrum of **1**. This is in near perfect agreement with a weighted integral sum (87.8%) calculated by considering the positions of five aromatic, one methine and three methyl protons of **1** relative to the frequency of the SHARPER signal and the frequency profile presented in Fig. 3a. The 25:1 SHARPER: $CH_3$ signal intensity ratio is fully explained by accounting for the $J$-splitting of the $CH_3$ signal, number of protons contributing to the two signals, efficiency of the signal collapse and the narrowing of the SHARPER singlet (Supplementary Note 4).

As described previously[8], the SNR of the SHARPER spectrum can be enhanced by a factor of 1.41, by removing the imaginary time domain data (Supplementary Note 5). Using matched filters to maximise independently the SNR[11] of both spectra, these were reprocessed using line-broadening of 0.11 and 0.59 Hz, respectively. After this treatment, a 11.4-fold higher SNR, as determined by comparing the SHARPER singlet and the $CH_3$ doublet, was obtained. Note that the magnetic field inhomogeneity can influence the SNR in standard 1D spectra, and hence the level of enhancement achieved. In contrast, the self-compensating nature of the SHARPER acquisition means that even severe magnetic field inhomogeneity has no or little effect on the intensity of the SHARPER singlet (Supplementary Note 6).

### Managing the power deposition to the probe

When executing pulse sequences containing tens of thousands of high-power pulses, an important experimental consideration is the power deposition. This is limited for liquid-state NMR probes, in particular cryoprobes (Supplementary Note 7). One way to reduce the deposited power, is to reduce the flip angle, α, of the spin-echo pulses. This may seem inefficient at first, as signal recovery[12,13] during a train of spin-

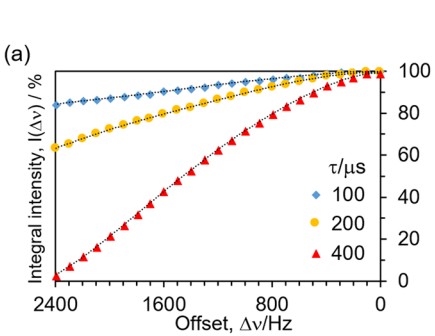

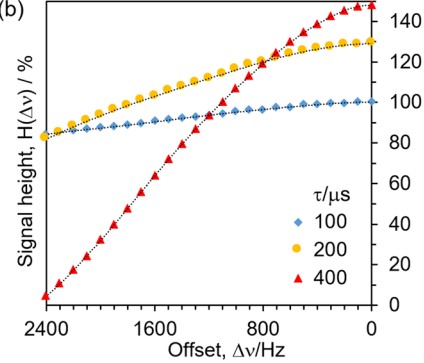

**Fig. 3 | Efficiency of the collapsing of NMR spectra into a singlet. a** Relative integral intensity, $I(\Delta\nu)/I(0, \tau = 100\mu s)$, and (**b**) relative signal height, $H(\Delta\nu) = T_2^S \times I(\Delta\nu)/(T_2^S(\tau = 100\mu s) \times I(0, \tau = 100\mu s))$, of the SHARPER singlet of HOD in doped $D_2O$ at 400 MHz as a function of the frequency offset, $\Delta\nu$. Data was normalised to the integral intensity (**a**) and the hight (**b**) of the on-resonance signal based on $\tau = 100$ μs. Data for chunk times $\tau = 100$ (blue), 200 (yellow) and 400 μs (red) are presented. Source data are provided as a Source Data Fig. 3.xlsx.

echoes over a 1/τ [Hz] frequency range scales with sin(α/2). For α = 90°, this means reduction to 71% relative to α = 180°. However, as demonstrated below, this loss of integral intensity is largely compensated for by narrowing of the SHARPER singlet, which originates from two sources. Firstly, the relative increase of the chunk time *vs* the pulse duration leads to longer $T_2^S$. Secondly, for 90° pulses, the spin-lattice relaxation contributes[14] towards $T_2^S$ and for molecules outside of the extreme narrowing limit ($T_1 > T_2$) this increases $T_2^S$ further. Both factor therefore contribute to the narrowing of the SHARPER signal and thus its increase intensity (Supplementary Note 7), while the power input into the probe is halved.

## Selecting signals to be collapsed

When measuring a whole-molecule property, such as a diffusion coefficient of a pure compound, the chemical shift resolution can be sacrificed; the same information is much more efficiently obtained from a narrow SHARPER singlet – an intense signature of a molecule. Nevertheless, care must be taken not to include signals that could compromise the outcome of such experiments. The signals to be excluded may include those of the solvent, labile protons, which can be in exchange with the protons of the solvent, or in case of organic solvents, with traces of water and a signal of the chemical shift reference standard. The SHARPER-DOSY experiment therefore typically starts with a suppression of selected signals, followed by a band-selective excitation of the spectral region destined to be collapsed.

This process is illustrated at 800 MHz on a sample of **2** in benzene-$d_6$ ($C_{62}H_{111}N_{11}O_1$, $M_w = 1{,}202.61$ g/mol), which contains a small amount of $H_2O$ in slow exchange with one OH and four NH protons in **2**. The $C_6HD_5$ signal at 7.2 ppm is particularly intense, while the $H_2O$/OH signals resonating at ~ 1.55 ppm are relatively weak. It is well known from protein NMR spectroscopy that the water signal is best pre-saturated when the carrier frequency is set to the $H_2O$ frequency. However, the nature of the SHARPER acquisition dictates for the r.f. carrier to be placed in the middle of the collapsed region, which for **2** is at 3.27 ppm, in the middle of aliphatic resonances. This positions the $C_6HD_5$ and OH signals off-resonance. Nevertheless, a protocol exists[15], which produces an optimal suppression of off-resonance signals. It involves the use of ~20-80 ms phase-ramped, low power rectangular pulses applied in a loop with the pulse length adjusted to allow a 2nπ (*n* is an integer) rotation for the off-resonance signal(s). With only one signal to be suppressed, this condition is easily fulfilled. We show here that for multiple signal suppression sites, a numerical solution can be found that allows each off-resonance signal to undergo close to a multiple number of full rotations, achieving optimal signal suppression (Supplementary Note 8). In the case of **2**, $^{13}C$ satellites of $C_6HD_5$

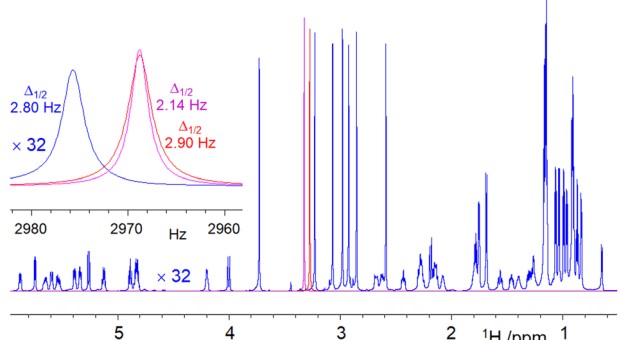

**Fig. 4 | Collapsing the displayed region of the $^1H$ NMR spectrum of 2.** Shown is an overlay of a partial 1D $^1H$ spectrum of **2** (blue) and two SHARPER spectra acquired using 90°(magenta, offset by +35 Hz for visibility) and 180°(red) $^1H$ spin-echo pulses. The vertical scale of the 1D spectrum was increased 32 times. The inset shows a close up of the two SHARPER singlets and the 3.72 ppm $NCH_3$ singlet. All spectra were produced using matched filters, hence the stated $\Delta_{1/2}$ values are double the original linewidths.

(Supplementary Note 9.1) were also saturated by applying a low power $^{13}C$ decoupling during the pre-saturation period.

In order to avoid the interference caused by the exchange of the NH protons of **2** with water, these were not included in the SHARPER signal. The remaining 106 protons were selected using a perfect echo[16,17], in which short 180° ReBurp[18] pulses surrounded by pulsed field gradients (PFGs) replaced the non-selective 180° spin-echo pulses (Supplementary Note 1.2). The efficiency of signal selection by this band-selective perfect echo (BSPE) was 90% (Supplementary Note 9.2). The BSPE- SHARPER spectra of **2** obtained with 180° (90°) spin-echo pulses, resulted in 97% (75%) signal recovery relative to the entire BSPE spectrum. An overlay of the three spectra processed with matched exponential filters and with imaginary time domain data removed for the SHARPER data, is shown in Fig. 4. In this presentation, the 1D spectrum was scaled up 32 times to achieve parity of the N-$CH_3$ singlets with the SHARPER singlets, revealing a 96-fold intensity increase of the SHARPER

signal relative to a hypothetical one proton singlet from a 1D spectrum – a remarkable sensitivity increase. The inset in Fig. 4 shows an expansion of the BSPE-SHARPER singlets obtained using 90° and 180° spin-echo pulses, respectively, and that of the $NCH_3$ signal at 3.722 ppm from the 1D spectrum. The SHARPER singlet obtained using 90°spin-echo pulses is the narrowest, compensating for the loss of

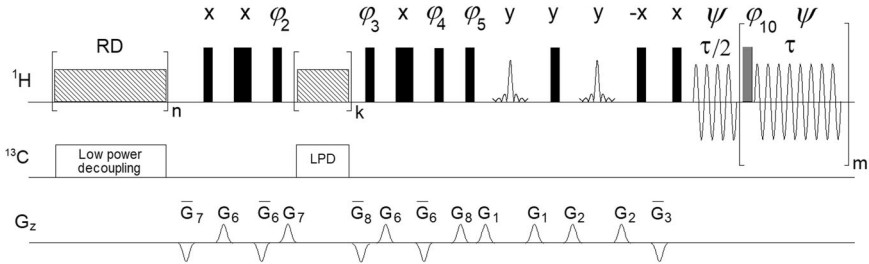

**Fig. 5 | Pulse sequence of SHARPER-DOSY.** For explanation of symbols and details see Supplementary Note 1.3.

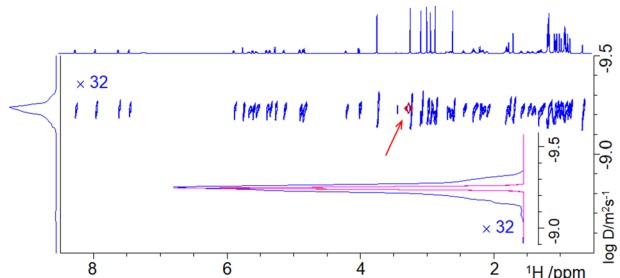

**Fig. 6 | Efficient determination of a diffusion coefficient.** An overlay of the DOSY (blue) and two SHARPER-DOSY spectra of **2** acquired using 90° (magenta) and 180° (red) spin-echo pulses, which overlay perfectly. Red arrow points to the SHARPER-DOSY cross peaks. The inset shows an overlay of the $F_1$ projection of the three spectra. The DOSY spectrum and its projection were scaled up 32 times.

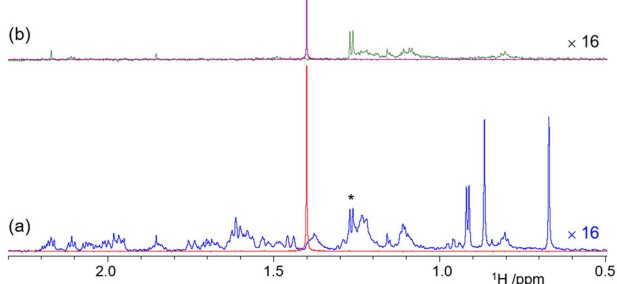

**Fig. 7 | Recognizing solvent impurities.** An overlay of 1D $^1$H (blue and green, 128 scans) and BSPE-SHARPER spectra (red and magenta, 8 scans) of (**a**) 1.3 µg of **3** and (**b**) a blank $D_2O$ sample. Protons 3, 7 and 12 (Fig. 1) were not included in the collapsed signal. The regular 1D spectra were scaled up 16 times and the doublet at 1.28 ppm is indicated by an asterisk.

integral intensity and surpassing the height of the SHARPER singlet acquired using 180° pulses.

### SHARPER- DOSY

Inserting a DOSY module[19,20] between the signal pre-saturation and the BSPE block, leads to a SHARPER-DOSY pulse sequence (Fig. 5). Applied to acquire two 2D SHARPER-DOSY spectra of **2** using 90° and 180° spin-echo pulses these are overlaid with a regular

DOSY spectrum of **2** in Fig. 6 (see also Supplementary Note 9.3). Also here a scaling factor of 32 was used for the regular DOSY spectrum, which shows a familiar smearing of signals in the DOSY dimension, while the SHARPER-DOSY spectra have much narrower $F_1$ profiles. All three diffusion axis projections overlap perfectly (inset in Fig. 6).

### SHARPER-DOSY using µg quantities of sample

Having demonstrated a flawless performance of SHARPER-DOSY, the technique was tested at 800 MHz on a microgram quantity of a medium size organic molecule, sodium cholate ($C_{24}H_{39}O_5Na$, $M_w = 430.55\,g/mol$) containing 1.3 µg of the compound in 0.55 ml of $D_2O$ ([**3**] = 5.5 µM). Three deshielded HO-CH protons 3, 7 and 12 (Fig. 1) of **3** were excluded and the remaining 33 protons resonating within a ± 0.75 ppm range were collapsed into a SHARPER singlet. Such narrow frequency range allowed the chunk time to be increased to $\tau = 448$ µs, achieving >80% signal recovery. An overlay of a 1D BSPE spectrum of **3** acquired using 128 scans with a 8-scan BSPE-SHARPER spectrum (180° spin-echo pulses, Fig. 7a) indicates that the SHARPER signal is 22-fold more intense than the $CH_3$ singlets of **3**. This is twice the 11-fold ratio calculated based on the proton count (33H/3H of a $CH_3$ singlet = 11). Narrowing of the SHARPER signal accounts for some of the observed excess gain, but does not explain it completely.

A closer inspection of the 1D BSPE spectrum of **3** revealed a doublet at 1.28 ppm that does not belong to this compound. Repeating the same experiments on a "pure" $D_2O$ sample showed that this and other signals are from solvent impurities (Fig. 7b). These account for the rest of the intensity of the SHARPER signal of **3** and pose a problem

for the acquisition of SHARPER-DOSY spectra. The collapsed signal is not solely from the compound of interest but contains the net and potentially significant contribution from a range of minor solvent impurities. A simple way around this problem is to collect two SHARPER-DOSY spectra; one of the sample and the other of the solvent, followed by a subtraction of the two 2D data sets (Bruker AU program, dosy_adsu, is provided in the Supplementary Software) before DOSY processing, as illustrated on 1D traces of the respective spectra in Fig. 8a. An overlay of $F_1$ projections of four DOSY spectra (Fig. 8b) shows that before this treatment, the projection of the SHARPER-DOSY spectrum of **3** was broad.

The projection of the $D_2O$ spectrum was yet broader, reflecting presence of several different size impurities. In contrast, the projection of the difference spectrum is narrower and much more intense than the projection of a regular DOSY spectrum, in which a contribution of the three $CH_3$ signals of **3** dominates the $F_1$ projection. With the experimental time of 10 min per a 2D SHARPER-DOSY spectrum, the diffusion coefficient of this medium size organic molecule was thus reliably determined using 1.3 µg of sample in 20 min on a 800 MHz cryoprobe NMR spectrometer.

The difference spectrum was used to compare the accuracy of the determination of a diffusion coefficient of low concentration compounds by analysing spectra or the time domain data. It was found that a more accurate value of a diffusion coefficient of **3** was obtained when time domain points up to 1.26 $T_2^S$ were integrated[21] as opposed to using all time domain points or spectral integrals. For details see Supplementary Note 10.

## Discussion

As demonstrate above, SHAREPR-DOSY technique provides significant sensitivity improvement for the measurement of diffusion coefficients of pure compounds. The experiment typically involves suppression of the residual $^1$H signals of deuterated solvents and may use a band selective excitation to exclude some resonance. For example signals of

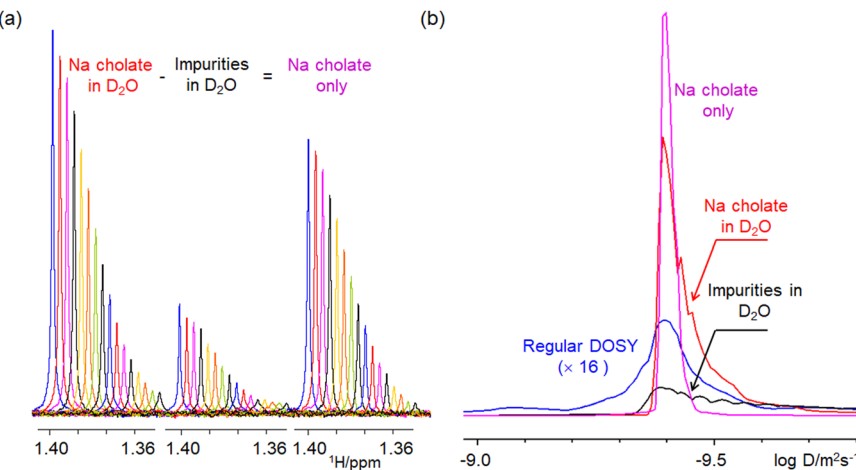

**Fig. 8 | Subtracting solvent impurities improves the quality of DOSY spectra.**
**a** SHARPER-DOSY signals from 16 spectra acquired by increasing the strength of pulsed field gradients for sample of **3**, sample of $D_2O$ and their difference; **b** overlay of the $F_1$ projections of the DOSY spectrum (blue), the SHARPER-DOSY spectra of **3** in $D_2O$ (red), $D_2O$ impurities (black), and of their difference spectrum (magenta).

labile protons, or isolated signals that would not contribute substantially to the SHARPER signal and their inclusion could invalidate the results. These two pulse sequence elements are also applicable when strong signals of co-solvents or buffers are present, potentially sacrificing more of the signal, focusing on narrower spectral regions only containing signals of the studied compound. Taking this approach to its limits, very selective techniques exist that can excite a single multiplet in a spectrum of a mixtures of compounds[22,23]. These can be implemented into the SHARPER-DOSY pulse sequence to achieve selectivity that is essential for accurate determination of diffusion coefficients of compounds in mixtures, while enhancing sensitivity by removing splittings and sharpening the acquired signal[5,8]. For mixtures of small and large compounds, preselection of small molecules can be achieved by including $T_2$ filters[24], while large molecules can be selected using diffusion filters[24], thus reducing the complexity of spectra. These techniques can be used e.g. to remove signals of co-solvents, buffers or small molecular impurities in samples of polymers. As exemplified in this work, solvent impurities can be dealt with by subtracting the SHARPER spectra of the solvent. Although only examples of ¹H NMR spectra are presented here, ¹⁹F spectra are equally amenable to SHARPER-DOSY methodology. While attention must be paid to a wider chemical shift range of ¹⁹F, the absence of a background signal due to lack of fluorinated endogenous compounds simplifies setting up of the DOSY experiments. Given the prevalence of manmade fluorinated drugs[25] and agrochemicals[26], together with a recent interest in polyfluoroalkyl substances[27], which typically are present at low concentrations, the SHARPER-DOSY technique is highly relevant to studies of fluorinated molecules.

The use of a CPMG acquisition loop is essential for NMR studies, typically carried out at low or very low magnetic fields, of a wide range of materials that affect the magnetic field homogeneity, including porous rocks[28], oil and gas[29], and dairy products[30]. A very short spin-echo times (0.1–0.5 ms) used in these experiments reduce primarily the effect of background gradients on decay of the NMR signal, while at the same time eliminate the chemical shift dispersion and $J$ coupling evolution. On high-field solution-state NMR spectrometers, signal broadening due to magnetic field inhomogeneity is typically small (sub Hz to few Hz), nevertheless, the chemical shift dispersion on the order of thousands of Hz needs to be eliminated. Addressing this requirement, $J$ evolution due HH couplings is therefore also suppressed[9], as removing chemical shifts typically requires the use of spin-echo times below 0.5 ms. It is worth noting, that although a narrower chemical shift range is efficiently collapsed by 90° spin-echo pulses, the

accompanying signal loss is compensated for by longer $T_2^S$ relaxation times, resulting in narrower, more intense signals, while the power deposition into probes is reduced.

A recent extension of Laplace NMR–the ultrafast multidimensional Laplace NMR[31–33],–enabled correlation of relaxation and diffusion parameters as well as the observation of molecular exchange phenomena in single scan experiments. These experiments typically use longer spin echo times (5–30 ms) in combination with spatial encoding and therefore exhibit reduced sensitivity. In their basic form they do not eliminate $J$ evolution; this can be achieved[32] by using perfect echos[16,17] in place of a CPMG sequence in the acquisition loop. Such pulse sequence element was not used in our work, as it would cause an undesirable line broadening of signals.

The CPMG acquisition module has also been utilised to enhance the sensitivity of solid-[34–38] and liquid-state[39–41] experiments, some applied in severally inhomogeneous magnetic fields[42–44]. Parallel efforts to increase the sensitivity of liquid-state NMR experiments such as manipulation of solvent exchange with labile sites in biomolecules[45], the use of nanoliter non-resonant coils[46], various forms of dynamic nuclear polarisation[47–49], exploitation of electronic spins in nitrogen-vacancy centres[50], or the use of *para*-hydrogen induced hyperpolarization (PHIP) based on chemical reactions[51] or reversible processes (SABRE)[52,53] are the focal point of many researcher groups. As demonstrated recently by the SABRE-SHARPER experiment[8], the concept of collapsing resonances in high-resolution NMR is compatible with hyperpolarisation techniques and can produce significant additional sensitivity gains by removing frequency modulation of the NMR signal and importantly, also counter the magnetic field inhomogeneity.

Measurement of diffusion coefficients of low-micromolar samples in comparable time to those achieved by SHARPER-DOSY has been so far only possible for compounds that are amenable to SABRE hyperpolarisation[54]. In comparison, single-scan DOSY measurements, which utilise spatial encoding, require much more concentrated samples[55,56].

In conclusion, the developed protocol for the measurement of diffusion coefficients of single compounds increased the sensitivity of standard DOSY experiments up to two orders of magnitude, with the exact gains depending on the number of collapsed resonance and their relaxation properties. Smaller molecules containing less protons, nevertheless have longer spin-spin relaxation times that narrow the SHARPER singlets (for comparison, $\triangle_{1/2}^S(\alpha = 180°) = 0.11$, 1.45 and 1.1 Hz for compounds **1**, **2** and **3**, respectively). Achieving the SNR gains

above those expected based on the proton count only is therefore possible. When used on high field cryoprobe NMR instruments, submicrogram quantities of compounds are sufficient to obtain high quality DOSY spectra. This e.g. allows to access low concentration samples in studies of molecular interactions and aggregation. As outlined in the Discussion, the SHARPER-DOSY technique can also be modified for applications to mixtures of varied complexity.

Due to the narrower frequency range, the relative gains at the lower magnetic fields, such as used in benchtop NMR instruments, will be higher than at very high fields. Finally, the concept of collapsing spectra or parts thereof by SHARPER acquisition as outlined here opens new possibilities for characterisation of molecules, or their mixtures, and is expected to find numerous applications in liquid-state NMR spectroscopy.

## Methods

**Doped water** A standard sample (0.1 mg $GdCl_3$/ml $D_2O$ + 1% $H_2O$ + 0.1% $^{13}CH_3OH$) was used. The data was acquired on and processed in TopSpin3.2 a 400 MHz Bruker AVANCE III spectrometer using the sharper_collapse.du pulse sequence (see Supplementary Software) and following parameters: 1 s relaxation time (D1), 1.229 s nominal acquisition time (AQ), 2 dummy (DS) and 2 real (NS) scans, 20,000 Hz (49.983 ppm) spectral width (SW), 49152 time domain points (TD), $pw_{90°} = 30 \mu s$, $pw_{180°} = 60 \mu s$ at 4.46 W, acquisition chunk times, τ, of 100, 200 and 400 μs, 25 μs dwell time (DW) with 4, 8 or 16 points per chunk, and 12,228, 6,144 or 3,072 spin echoes, respectively. The $T_1$ and $T_2$ relaxation time determined by inversion recovery and a CPMG methods were 230 and 180 ms, respectively.

**Compound 1** (40 μL in 550 μL of $D_2O$, c = 0.562 M). The 1D $^1H$ NMR spectrum was acquired and processed in TopSpin3.2 on a 400 MHz Bruker AVANCE III spectrometer using the following parameters: D1 = 16 s, AQ = 9.83 s, DS = 2, NS = 2, SW = 10,000 Hz (49.983 ppm), TD = 196608. The 1D SHARPER spectrum was acquired using the sharper_collapse.du pulse sequence and identical common parameters as for the 1D $^1H$ spectrum. The following specific parameters were used: τ = 200 μs, $pw_{90°} = 30 \mu s$, $pw_{180°} = 60 \mu s$ at 4.46 W, DW = 50 μs (4 points per chunk, 49152 spin echoes). The actual acquisition time was $AQ*(\tau + pw_{180°})/\tau = 12.78s$.

**Compound 2** ($M_w = 1,202.61$ g/mol, 38 mg in 550 μL of benzene-$d_6$, c = 3.67 mM). The data was acquired and processed in TopSpin4.1 on an 800 MHz BRUKER NEO NMR spectrometer equipped with a TCI cryoprobe. For all experiments, identical common parameters, as stated for the 1D $^1H$ spectrum, were used: D1 = 3.0 s, AQ = 1.652 s, DS = 4, NS = 8, SW = 39682.54 Hz (49.6384 ppm), TD = 128k. Presaturation parameters: pw = 25764.46 μs, l6 = 117, carrier frequency o1 = 2612.50 Hz, $v(C_6HD_5) = (o1 + 3105.05)$ Hz; $v(H_2O) = (o1 - 2173.57)$ Hz (optimised primarily for the suppression of $C_6HD_5$). Pulse sequence zgpr_pulse.du (see Supplementary Software). For $^{13}C$ decoupling a xy32 super cycle[57,58] modified to implement composite 180° pulses ($90^o_x 180^o_y 90^o_x$) with $pw_{90°} = 192 \mu s$ was used with the $^{13}C$ carrier frequency at 128 ppm.

BSPE spectrum (pulse sequence zgpebs.du, see Supplementary Software) and the BSPE-SHARPER spectra (sharper_collapse.du) the following parameters were used: 1 ms ReBurp pulse, 600 μs PFG, $G_1 = 7\%$, $G_2 = 5\%$ and $G_3 = -12\%$. For the BSPE-SHARPER experiments the specific parameters were: τ = 100.8 μs, DW = 12.6 μs (8 points per chunk, 16,384 spin echoes), the spin echo pulses, $pw_{90°} = 40 \mu s$, $pw_{180°} = 80 \mu s$ at 0.33 W resulting in the actual acquisition time $=AQ*(\tau + pw_{180°})/\tau = 2.96s$ and $=AQ*(\tau + pw_{90°})/\tau = 2.30s$. Spectra were processed using matched filters, line broadening LB = 1.4 Hz for the 1D and BSPE spectra and LB = 1.45 and 1.07 Hz for BSPE-SHARPER with 180° and 90° spin-echo pulses, respectively.

A reference 2D DOSY spectrum (pulse sequence ledbpgp2s.compensated.dn, see Supplementary Software) was acquired using a modified Bruker pulse sequence, *ledbpgp2s*, to include a

compensating PFGs before the start of the pulse sequence and off-resonance presaturation as explained above. The following DOSY specific parameters were used: diffusion time, d20 = 200 ms, diffusion PFGs, p30 = 1 ms, the spoil and compensation PFGs, p19 = 0.6 ms and the eddy current delay d21 = 5 ms. All PFGs were sine shaped and applied at the strength specified in the pulse programme. The diffusion gradients were ramped up in 16 increments using 5 to 95 % strength of the PFG coil (66.4 G/cm). Number of scans was 8 per increment, yielding total acquisition time of 11 min. The SHARPER-DOSY spectra (pulse sequence ledbpgp2s.sharper_collapse.du, see Supplementary Software) were acquired using the combination of parameters used for the BSPE-SHARPER and DOSY experiments stated above. The overall acquisition time was 14 and 12.5 min for the 180 and 90° spin echo pulses. The spectra were processed using matched filters, line broadening LB = 1.56 Hz for the 2D DOSY spectrum and LB = 1.39 and 1.02 Hz for SHARPER-DOSY with 180° or 90° spin-echo pulses, respectively. The number of points in the $F_1$ was 256, linear prediction was not used.

**Compound 3** ($M_w = 430.55$ g/mol, Sample 1: c = 5.5 μM, 1.3 μg; Sample 2: 7.7 mM, 1.82 mg in 550 μL of $D_2O$). The data were acquired and processed in TopSpin4.1 on an 800 MHz BRUKER NEO NMR spectrometer equipped with a TCI cryoprobe. For all experiments, identical common parameters, as stated for the 1D $^1H$ spectrum (zgpr_pulse.du), were used: D1 = 3.0 s, AQ = 1.05 s, DS = 4, NS = 128, SW = 15625 Hz (19.5451 ppm), TD = 32 k. The HOD signal presaturation was performed using the PRESAT_JUMP option with $\gamma B_1/2\pi = 96$ Hz.

For the acquisition of the BSPE spectrum (zgpebs.du) and the BSPE-SHARPER spectra (sharper_collapse.du) the following parameters were used: 3 ms ReBurp pulse, 600 μs PFG, $G_1 = 7\%$, $G_2 = 5\%$ and $G_3 = 12\%$. For the BSPE-SHARPER experiments the specific parameters were: τ = 448 μs, DW = 32 μs (14 points per chunk, 2340 spin echoes), the spin echo pulses, $pw_{90°} = 15.7 \mu s$, $pw_{180°} = 31.3 \mu s$ at 2.78 W resulting in the actual acquisition time $= AQ*(\tau + pw_{180°})/\tau = 1.12s$.

The spectra were processed using matched exponential filters with broadening, LB = 1.5 and 0.8 Hz (BSPE and BSPE-SHARPER spectrum of the **3**) and LB = 1.25 or 0.53 Hz (BSPE and BSPE-SHARPER spectrum of $D_2O$ impurities).

A reference 2D DOSY spectrum (ledbpgp2s.compensated.dn) was acquired using a modified Bruker pulse sequence, *ledbpgp2s*, to include a compensating PFGs before the start of the pulse sequence and off-resonance presaturation. The following DOSY specific parameters were used: diffusion time, d20 = 200 ms, diffusion PFGs, p30 = 1 ms, the spoil and compensation PFGs, p19 = 0.6 ms and the eddy current delay d21 = 5 ms. All PFGs were sine shaped and applied at the strength specified in the pulse programme. The diffusion gradients were ramped up in 16 increments using 5 to 95 % strength of the PFG coil (66.4 G/cm). Number of scans was 8 per increment, yielding total acquisition time of 10 min. The SHARPER-DOSY spectra (ledbpgp2s.sharper_collapse.du) were acquired using the combination of parameters used for the BSPE-SHARPER and DOSY experiments stated above.

## Data availability

The NMR Data, Supplementary Software and Source Data Files generated in this study have been deposited in Edinburgh DataShare database[59], under accession code https://doi.org/10.7488/ds/7472. Till August 31, 2023 the data are available from the corresponding author and freely available after this date. Source data are provided with this paper.

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

## Acknowledgements

This research was supported by EPSRC grants EP/SO16139/1 (DU) and EP/R030065/1 (DU). We thank Megan Halse, University of York, for valuable discussion during the development of the project.

## Author contributions

D.U., C.L.D. and G.P. contributed to the design of experiments. Spectra were acquired and analysed by D.U., G.P. and P.J.B.. P.J.B. has written the programmes for optimisation of signal suppression and removal of the imaginary time domain data. G.P. has written the A.U. programme for the subtraction of 2D DOSY data sets. D.U. has written the the first version of the manuscript. G.C.L.-J. contributed to the writing of the manuscript.

## Competing interests

The authors declare no competing interests.
