## [Peer Review File · Nature Communications]

REVIEWER COMMENTS

Reviewer #1 (Remarks to the Author):

In this manuscript, Uhrin et al report a method to improve the sensitivity of NMR diffusion measurements of single compounds by one to two orders of magnitude by removing chemical shifts and J-couplings. The sensitivity enhancement arises from the SHARPER detection, previously published by the group of Uhrin (Ref. 5), in which all signals of the signals of the compound are merged to a single, narrower line by CPMG acquisition with a very short echo time (< 0.5 ms). In this manuscript, the authors combine the SHARPER sequence with the DOSY diffusion block to form a method called SHARPER-DOSY. Signals of other substances and solvent are removed by suppression and band selective excitation. First, the authors show the capability of the SHARPER experiment to collapse the signal of 1-phenylethanol into a single, tall and narrow signal. Then they use a doped water sample to demonstrate removal of frequency modulation over thousands of Hz. Finally, they demonstrate feasibility of the SHARPER-DOSY to measure diffusion with high sensitivity selectively for single compounds. Supplementary information includes pulse sequences as well as some Python scripts for data analysis. This is a very well-written manuscript and systematic study, and the proposed technique allows investigation of diffusion of low concentration substances.

Modification suggestions:

It would be useful to have examples of raw data in a figure in supporting material, with an explanation, how data is manipulated before Fourier transform. How gaps in data acquisition due to 180 (or 90) degree pulses are filled or are they just removed?

In principle, Fourier transform is not needed as there is no frequency resolution in the SHARPER-DOSY data, but signal intensity could be quantified by integrating time domain data. Naturally FT is a nice way for quantitative and visual comparison with conventional DOSY data. Still, it would be interesting to see a short discussion of the advantages and disadvantages of these two processing/quantification approaches, e.g., in terms of SNR.

In addition to the masses of the analyzed samples, it would be useful to see concentrations.

Check lines 49-53 on page 2 (unpaired parentheses, misspelled "bellow").

It is not advisable to combine red and green spectra in a figure because of potential color-blind readers.

Caption of Fig. 7: "filed" -> "field".

Caption of Fig. S1.2: sentence beginning with "The phases of the BSPE experiment are..." appears twice.

Reviewer #2 (Remarks to the Author):

Nature-comm:

SHARPER-DOSY: Sensitivity Enhanced Diffusion-Ordered NMR Spectroscopy

by George Peat et al.

This paper presents an NMR technique (combining the previously published SHARPER technique and DOSY) to achieve sensitivity enhancement. This enhancement is obtained by sacrificing the chemical shift and J-coupling resolution. The authors argue that since the diffusion coefficient of all nuclei of the same molecule would be identical, thus chemical shift resolution is not necessary. This is certainly a legitimate argument for the case of pure samples, however, it may severely limit the application of the method.

The majority of chemical/biological samples are a mixture of many components. Per-deuterated solvents are widely available, however, many co-solvents, buffers, and polymers that are essential for the chemical state of the sample may not be easily deuterated. In particular, if the targeted analyte is of very low concentration, the impurity hydrogens could often overwhelm the signal of the analyte. It would be very useful to discuss this limitation and perhaps potential solutions based on frequency selection or other methods.

The key part of the SHARPER technique use CPMG-like sequence with 180-degree pulses to refocus. It is well known that short echo times will stop the J evolution and thus remove J splitting (please add reference). The paper mentions the use of CPMG acquisition loop in other work (ref 20-25). It would be useful to contrast this current method with those in Ref 20-25 since the CPMG segment used in those studies are also very short and removed J-modulation and chemical shift.

REVIEWER COMMENTS

Reviewer #1 (Remarks to the Author):

In this manuscript, Uhrin et al report a method to improve the sensitivity of NMR diffusion measurements of single compounds by one to two orders of magnitude by removing chemical shifts and J-couplings. The sensitivity enhancement arises from the SHARPER detection, previously published by the group of Uhrin (Ref. 5), in which all signals of the signals of the compound are merged to a single, narrower line by CPMG acquisition with a very short echo time (< 0.5 ms). In this manuscript, the authors combine the SHARPER sequence with the DOSY diffusion block to form a method called SHARPER-DOSY. Signals of other substances and solvent are removed by suppression and band selective excitation. First, the authors show the capability of the SHARPER experiment to collapse the signal of 1-phenylethanol into a single, tall and narrow signal. Then they use a doped water sample to demonstrate removal of frequency modulation over thousands of Hz. Finally, they demonstrate feasibility of the SHARPER-DOSY to measure diffusion with high sensitivity selectively for single compounds. Supplementary information includes pulse sequences as well as some Python scripts for data analysis. This is a very well-written manuscript and systematic study, and the proposed technique allows investigation of diffusion of low concentration substances.

We thank the reviewer for the summary and a positive evaluation of our write up.

Modification suggestions:

It would be useful to have examples of raw data in a figure in supporting material, with an explanation, how data is manipulated before Fourier transform. How gaps in data acquisition due to 180 (or 90) degree pulses are filled or are they just removed?

We have included the raw data (SHARPER time domain data) of 1-phenylethanol, 1, in the Supplementary Data and analysed them together with the corresponding spectra. The following content was added:

The time domain data of experiments in which the signal is manipulated during acquisition is acquired effortlessly by Bruker Topspin pulse programs. Data acquisition is paused at the end of each data chunk and restarted after a block of pulses and delays. The build-up of the signal can be viewed in real time on a computer screen. No data manipulation is required, chunks of data points are arranged to form an "FID" that can be processed in a regular manner in Topspin or any other NMR package.

The time domain data points and spectra of **1** presented in Fig. 1 of the main paper are analysed in more detail here. Figures S2.1a and b show the real and imaginary parts of the time domain data acquired using the pulse sequence of Fig. 1b. In this experiment, the decaying signal was directed into the real channel by adjusting the phase of the receiver. The inset in Fig. S2.1a shows the first 200 real points acquired over 20 ms with the initial group delay points of the Bruker Avance digital filter removed (these points do not contain any spectral information). The points acquired during the first few echoes show intensity fluctuations, which settle quickly.¹ A slow modulation of the time domain points with a period of 0.384s visible in both channels remains unexplained; nevertheless, it did not manifest itself in spectra shown in Fig. S2.1. d-g.

The spectra were obtained by including data from both channels (c and e) and by using only the real data points (d, f and g). The latter spectra show reduced noise over the entire frequency range, and in particular around the central peak (see insets in c and d). The intensity of the chunking sidebands that occur just below ± 5000 Hz ($= 1/\tau = 1/(200e-6)$ Hz) is reduced when only the real data are used; their intensity is at ~ 3% of the main SHARPER peak (see insets in e and f). The resonance frequency of the sidebands is slightly lower than 5000 Hz due the additional delays of the order of

nanoseconds, which are inserted into the spin-echoes to allow switching of the receiver on and off. The low level artefacts in the \pm (2000-3000) Hz regions cannot be fully explained; their broader components can be removed by applying a backward linear prediction (see g). Overall, the SHARPER spectra of **1** are very clean; the intensity of the described minor signals is low and inconsequential for the quantification of the SHARPER signal, which sits in the clean central region of the spectrum.

Fig. S2.1. 400 MHz ^1H SHARPER data presented in Fig. 1 of the main paper. (a) Real time domain data points. The inset shows the first 200 points acquired over 20 ms. (b) Imaginary time domain points. (c) Spectra obtained using both real and imaginary points and line broadening of 0.11 Hz (a matched filter). (d) The same as (c) but using the real data points only. The insets show vertical expansions of the SHARPER signal over ± 1 and ± 40 Hz, respectively. (e) and (f) present a 1024-fold vertical expansion of (c) and (d). The insets in (e) and (f) show 8-fold vertical expansions of the areas around the first sidebands relative to (c) and (d). (g) The same as (f) but with a linear backward prediction of the first 100 points applied to real time domain points only. For comments, see the text.

In principle, Fourier transform is not needed as there is no frequency resolution in the SHARPER-DOSY data, but signal intensity could be quantified by integrating time domain data. Naturally FT is a nice way for quantitative and visual comparison with conventional DOSY data. Still, it would be interesting to see a short discussion of the advantages and disadvantages of these two processing/quantification approaches, e.g., in terms of SNR.

We would like to thank the referee for this comment. Our analysis showed that for noise spectra a judicial interpretation of the time domain data yields more accurate values of diffusion coefficients than analysis of spectra.

The following text was added to main the manuscript:

The difference spectrum was used to compare the accuracy of the determination of a diffusion coefficient of low concentration compounds by two methods; analysis based on spectra and the time domain data. It was

found that more accurate value of a diffusion coefficient of **3** was obtained when time the domain points up to $1.26 T_2^S$ were integrated²¹ as opposed to using all time domain points or spectral integrals. For details see Supplementary Data 11.

The following text was added to the Supplementary Data:

The diffusion coefficients of sodium cholate in D₂O were evaluated at two different concentrations by analysing spectra (Fig. 7 for the dilute sample, spectra for the concentrated sample are not shown) and the time domain points of a series DOSY spectra acquired using increasing gradient strength (Fig. S11.1).

Fig. S11.1. The time domain data of 16 DOSY spectra acquired on (a) 7.7 mM and (b) 5.5 μM sample of sodium cholate, **3**. The two figures are not to scale; (b) presents the time domain data after the subtraction of water impurities.

The results are summarised in Table S1.

Table S1. Diffusion coefficient of sodium cholate, **3**, at two concentrations and two methods of evaluation.

	Spectra integrated ^(a)	Time domain integrated ^(d)	
Concentration	Matched exponential line-broadening + FT	0-16,384 real points	Up to $1.26 T_2^S$ ^(e)
	Diffusion coefficient $D \times 10^9 / \text{m}^2 \text{s}^{-1}$		
7.7 mM	0.353 ± 0.001 ^(b)	0.353 ± 0.001	0.354 ± 0.001
5.5 μM	0.407 ± 0.003 ^(c)	0.407 ± 0.003	0.387 ± 0.003

^a All 16k of real points were zero filled to 128k points and a matched exponential line broadening applied (LB = 1.1 Hz and 0.78 Hz for the concentrated and dilute sample, respectively). Fifteen 1D DOSY spectra acquired as detailed in the Methods section of the main paper were fitted in Excel using Eqn 14.

$$\ln(I) = (\gamma_{^1\text{H}} g \delta)^2 \left(\Delta - \frac{\delta}{3} \right) D \quad (14)$$

where I is the integral intensity, either of the SHARPER spectra or the time domain points, $\gamma_{^1\text{H}}$ is the gyromagnetic ratio of proton, g is the gradient amplitude, δ is the gradient length, Δ is the diffusion delay and D is the diffusion coefficient.

^b Individual spectra had a signal-to-noise ratio (SNR) from 149,311:1 (1st spectrum) to 9636:1 (15th spectrum) and were integrated over the region of ± 2400 Hz to include the base of the SHARPER peak.

^c Individual spectra had SNR from 135:1 (1st spectrum) to 6:1 (15th spectrum) and were integrated over the region of ± 16 Hz to include the base of the SHARPER peak.

^d Bruker time domain data were extracted using NMRglue (<https://www.nmr glue.com/>) and u to 16k of real data integrated without any pre-processing by summing up their intensity.

^e As the maximum SNR of the time domain data is obtained³ for the acquisition time $1.26 T_2^S$, the integration of the time domain data was stopped at this value. For the concentrated and the dilute sample this represented 6000 and 8000 points, respectively.

As seen in Table S1, a diffusion coefficient, $D_{7.7 \text{ mM}} = 0.353 \pm 0.001 \times 10^9 / \text{m}^2 \text{s}^{-1}$, was obtained for the concentrated sample regardless of the data evaluation method. For the dilute sample, an identical diffusion coefficient, $D_{5.5 \text{ μM}} = 0.407 \pm 0.003 \times 10^9 / \text{m}^2 \text{s}^{-1}$, was obtained when spectra or time domain data points were integrated. However, the diffusion coefficient was different, $D_{5.5 \text{ μM}} = 0.387 \pm 0.003 \times 10^9 / \text{m}^2 \text{s}^{-1}$, when the points were truncated at the $1.26 T_2^S$.

To rationalise these results, our data were compared to those of Lindman *et al.*⁴ The authors measured diffusion coefficients of the cholate anion in H₂O by an open-ended capillary tube method Anderson and Saddington⁵ for a range of concentrations. Their data shows a linear dependency of the diffusion coefficient on concentration for solutions between

0.96 and 54 mM. After adjusting their diffusion coefficients for D₂O as the solvent used in our experiments ($D_{D_2O} = D_{H_2O}/1.25$),⁶ the following dependency of the diffusion coefficient on concentration was obtained:

$$D_A = -(0.0036 \pm 0.002)c + 0.356 \pm 0.006 \quad (15)$$

Values calculate using Eqn. 15 and our concentrations were then compared with the values obtained in this work (Table 2).

Table S2. Comparison of measured diffusion coefficients of **3** with the literature data.⁴

Conc.	$D_A \times 10^9 / \text{m}^2\text{s}^{-1}$ Based on Eqn. 15	$D_B \times 10^9 / \text{m}^2\text{s}^{-1}$ All time domain points, or spectra	$D_C \times 10^9 / \text{m}^2\text{s}^{-1}$ Timd domain points up to $1.26 T_2^S$	$\frac{D_A}{D_B}$	$\frac{D_A}{D_C}$
7.7 mM	0.333 ± 0.007	0.353 ± 0.001	0.353 ± 0.001	0.941	0.941
5.5 μM	0.360 ± 0.006	0.407 ± 0.003	0.387 ± 0.003	0.886	0.930

For a 7.7 mM sample, this comparison shows a small difference (5.9 %) between the absolute values of the diffusion coefficients obtained by the open capillary method and NMR, the latter being independent of the data evaluation method. For the 5.5 μM sample this difference raised marginally to 7% when the time domain was truncated at $1.26 T_2^S$. However, when spectra or all the time domain points were used to analyse the dilute sample data set, the difference increased to 11.4%.

The absolute differences between the open capillary and NMR methods can likely be attributed to systematic errors within either of the techniques. Nevertheless, the presented comparison indicates that for dilute samples processing of the initial time domain points leads to more accurate results than processing of all data or spectra.

The observed parity between the results obtained from spectra or all time domain data is not surprising as the two data sets are related by a mathematical operation, however, it should be noted that a matched exponential line broadening was only applied to spectra and not the time domain points. When integrating spectra, the results can be affected by noise within the integrated region, these therefore should be chosen carefully. Other possible processing approaches, i.e. apodising the time domain points prior to integration or applying stronger exponential line broadening prior to Fourier transformation of the data truncated at $1.26 T_2^S$ to avoid truncation artefacts were not explored..

In conclusions, based on this example, integration of real time domain points up to $1.26 T_2^S$ is recommended as an accurate method for the determination of diffusion coefficients from DOSY data of dilute samples.

In addition to the masses of the analyzed samples, it would be useful to see concentrations.

Three model compounds were used. Two had their concentration stated (Methods of the main paper) The concentration of 1-phenylethanol, **1**, was now added as well.

Check lines 49-53 on page 2 (unpaired parentheses, misspelled “bellow”).

These typos were corrected

It is not advisable to combine red and green spectra in a figure because of potential color-blind readers.

Our mistake, the colour scheme was changed.

Caption of Fig. 7: “filed” -> “field”.

This typos was corrected

Caption of Fig. S1.2: sentence beginning with “The phases of the BSPE experiment are...” appears twice.

This repetition was removed.

Reviewer #2 (Remarks to the Author):

Nature-comm:

SHARPER-DOSY: Sensitivity Enhanced Diffusion-Ordered NMR Spectroscopy

by George Peat et al.

This paper presents an NMR technique (combining the previously published SHARPER technique and DOSY) to achieve sensitivity enhancement. This enhancement is obtained by sacrificing the chemical shift and J-coupling resolution. The authors argue that since the diffusion coefficient of all nuclei of the same molecule would be identical, thus chemical shift resolution is not necessary. This is certainly a legitimate argument for the case of pure samples, however, it may severely limit the application of the method.

The majority of chemical/biological samples are a mixture of many components. Per-deuterated solvents are widely available, however, many co-solvents, buffers, and polymers that are essential for the chemical state of the sample may not be easily deuterated. In particular, if the targeted analyte is of very low concentration, the impurity hydrogens could often overwhelm the signal of the analyte. It would be very useful to discuss this limitation and perhaps potential solutions based on frequency selection or other methods.

Indeed, the unwanted signals need to be suppressed as described in the submitted manuscript (Subchapter: Selecting signals to be collapsed, page 5) even for pure molecules. We now discuss additional approaches, which increase the robustness of the technique.

The following text was added to the Discussion

As demonstrate above, SHARPER-DOSY technique provides significant sensitivity improvement for the measurement of diffusion coefficients of pure compounds. The experiment typically involves suppression of the residual ^1H signals of deuterated solvents and may use a band selective excitation to exclude some resonance, e.g. signals of labile protons, or isolated signals that would not contribute substantially to the SHARPER signal, while their inclusion would worsen the overall performance of the technique. These two pulse sequence elements are also applicable when strong signals of co-solvents or buffers are present, potentially sacrificing more of the signal, focusing on narrower spectral regions only containing signals of the studied compound. Taking this approach to its limits, very selective techniques exist that can excite a single multiplet in a spectrum of a mixtures of compounds.^{22,23} These can be implemented into the SHARPER-DOSY pulse sequence to achieve selectivity that is essential for accurate determination of diffusion coefficients of compounds in mixtures, while enhancing sensitivity by removing splittings and sharpening the acquired signal.^{5,8} For mixtures of small and large compounds, preselection of small molecules can be achieved by including T_2 filters,²⁴ while large molecules can be selected using diffusion filters,²⁴ thus reducing the complexity of spectra. These techniques can be used e.g. to remove signals of co-solvents, buffers or small molecular impurities in samples of polymers. As exemplified in this work, solvent impurities can be dealt with by subtracting the SHARPER spectra of the solvent. Although only examples of ^1H NMR spectra are presented here, ^{19}F spectra are equally amenable to SHARPER-DOSY methodology. While attention must be paid to a wider chemical shift range of ^{19}F , the absence of a background signal due to lack of fluorinated endogenous compounds simplifies setting up of the DOSY experiments. Given the prevalence of manmade fluorinated drugs²⁵ and agrochemicals,²⁶ together with a recent interest in polyfluoroalkyl substances,²⁷ which typically are present at low concentrations, the SHARPER-DOSY technique is highly relevant to studies of fluorinated molecules.

The key part of the SHARPER technique use CPMG-like sequence with 180-degree pulses to refocus.

It is well known that short echo times will stop the J evolution and thus remove J splitting (please add reference).

Reference added: Allerhand, A. Analysis of Carr—Purcell spin - echo nmr experiments on multiple - spin systems. I. The effect of homonuclear coupling. *The Journal of Chemical Physics* **44**, 1-9 (1966).

The paper mentions the use of CPMG acquisition loop in other work (ref 20-25). It would be useful to contrast this current method with those in Ref 20-25 since the CPMG segment used in those studies are also very short and removed J-modulation and chemical shift.

The following text was added to the Discussion.

The use of a CPMG acquisition loop is essential for NMR studies, typically carried out at low or very low magnetic fields, of a wide range of materials that affect the magnetic field homogeneity, including porous rocks,²⁸ oil and gas,²⁹ and dairy products.³⁰ A very short spin-echo times (0.1-0.5ms) used in these experiments reduce primarily the effect of background gradients on decay of the NMR signal, while at the same time eliminate the chemical shift dispersion and *J* coupling evolution. On high-field solution-state NMR spectrometers, signal broadening due to magnetic field inhomogeneity is typically small (sub Hz to few Hz), nevertheless, the chemical shift dispersion on the order of thousands of Hz needs to be eliminated. Addressing this requirement, *J* evolution due HH couplings is therefore also suppressed⁹, as removing chemical shifts typically requires the use of spin-echo times below 0.5 ms. Although a narrower chemical shift range can be efficiently collapsed by 90° spin-echo pulses, the accompanying signal loss is compensated for by longer T_2^S relaxation times, resulting in narrower, more intense signals.

REVIEWER COMMENTS

Reviewer #1 (Remarks to the Author):

I thank the authors for thorough and good answers for comments. However, I would like to have a clarification related to the answer to my first comment:

Please label the horizontal axis in Figs. S2.1 a and b and give units. Give also the dwell time (time between successive points), length of tau delay, and number of points collected during the tau delay. Why there is no signal interruptions due to pulses visible in the figures? The pulse length (60 us?) is probably long as compared to the dwell time (25 us), so the interruptions should be visible when zooming in to 100 us scale. Or are they just neglected?

Reviewer #2 (Remarks to the Author):

I am satisfied with the revision and I support the publication of this paper as is. Good work!

Reviewer #1 (Remarks to the Author):

I thank the authors for thorough and good answers for comments. However, I would like to have a clarification related to the answer to my first comment:

Please label the horizontal axis in Figs. S2.1 a and b and give units.

Our response: Apologies for this oversight. The units, seconds and milliseconds, respectively, for the spectra in Figs. S2.1a, b and the inset, were in the original figures but became masked during pasting from a PowerPoint into the Word document.

Give also the dwell time (time between successive points), length of tau delay, and number of points collected during the tau delay.

Our response: The parameters requested were included in the Methods section for compound **1** of the main manuscript and are now also given in the SI. The following parameters were used: dwell time = 50 μ s, chunk time, τ = 200 μ s, 4 points per chunk (or 2 complex points) and $pW_{180^\circ} = 60\mu$ s.

Why there is no signal interruptions due to pulses visible in the figures? The pulse length (60 μ s?) is probably long as compared to the dwell time (25 μ s), so the interruptions should be visible when zooming in to 100 μ s scale. Or are they just neglected?

Our response: In short, we have split the Fig. S2.1 into two and added a figure showing an expansion for 1ms of the time domain points. Here, the expected modulation with the frequency equal to inverse of the chunk time (and producing sidebands at $\pm 1/\tau$ ($1/200\text{e-}6$) = ± 5000 Hz) is visible. However, the inclusion of tens of μ s long nonselective pulses did not induce a step change in the intensity of the time domain points between individual acquisition chunks, as the pulses are many orders of magnitude shorter than the effective spin-spin relaxation time of **1** (2.89 s), which acts during the SHARPER acquisition. Nevertheless these pulses, contribute towards the overall T_2^S relaxation, shortening it and broadening (slightly) the SHARPER singlet.

To answer this and the previous points, Supplementary Data 2 was modified as follows:

2. Analysis of the time domain data and spectra of **1**

The time domain points of experiments, in which the signal is manipulated during acquisition, are acquired effortlessly by Bruker Topspin pulse programs. Data acquisition is paused at the end of each data chunk and restarted after a block of pulses and delays. The build-up of the signal can be viewed in real time on a computer screen. No data manipulation is required, chunks of data points are arranged to form an "FID" that can be processed in a regular manner in Topspin or any other NMR package.

The time domain data corresponding to the spectrum of **1** presented in Fig. 1 of the main paper is analysed here in more detail. Figures S2.1a and b show the real and imaginary parts time domain points acquired using the pulse sequence of Fig. 1b, dwell time of 50 μ s and 4 data points (i. e. 2 complex points) during a 200 μ s chunk time, τ . The decaying signal was directed into the real channel by adjusting the phase of the receiver. A slow modulation of the time domain points with a period of 0.384s visible in both channels remains unexplained; nevertheless, it did not manifest itself in the corresponding spectra shown in Fig. S2.2. Fig. S2.1c shows the first 200 real points acquired over 20 ms with the initial group delay points of the Bruker Avance digital filter removed (these points do not contain any spectral information). The

points acquired during the first few echoes show intensity fluctuations, which settle quickly.¹ Fig. S2.1d shows an expansion containing 10 real points acquired over a period of 1 ms. These show the expected modulation with the frequency equal to the inverse of the chunk time producing sidebands at $\pm 1/\tau$ ($1/200\text{e-6}$) = ± 5000 Hz. No intensity variations that could be attributed to the insertion of $60\mu\text{s}$ ^1H pulses between individual data chunks are visible. This is due to long effective relaxation of **1** during the SHARPER acquisition ($T_2^S > 2.89$ s, calculated as $T_2^S = 1/(\pi\Delta_{1/2})$, where $\Delta_{1/2} = 0.11$ Hz). A step decrease in the signal intensity was visible when selective pulses of comparable length to the long chunk times (~ 20 ms) were used to refocus the evolution of homonuclear coupling constants,² especially for fast relaxing spins of polymers. Nevertheless, as these discontinuities coincide with the chunk length, the artefacts they create in spectra appear at the same frequency as the chunking artefacts and do not interfere with the main signal (data not shown). The short nonselective r.f. pulses in experiments with chunk times of comparable length nevertheless contribute towards the overall T_2^S relaxation, shortening it and broadening (slightly) the SHARPER singlet.

Fig. S2.1. 400 MHz ^1H SHARPER time domain data points of the spectrum of **1** presented in Fig. 1 of the main paper; (a) and (b) show the real and imaginary points, respectively. (c) The first 200 real points acquired over 20 ms. (d) 10 real time domain points acquired over 1 ms. Modulation of the data over one chunk period of $200\mu\text{s}$ is highlighted. The following parameters were used: dwell time: $50\mu\text{s}$, chunk time: $\tau = 200\mu\text{s}$, and 4 data points (or 2 complex points).

The corresponding spectra obtained by Fourier transformation of the data points from both channels and by using only the real data points are shown in Fig. S2.2a, b and c, d, e, respectively. The latter spectra show reduced noise over the entire frequency range and in particular around the central peak (compare the insets on the right in Fig. S2.2a and c). The intensity of the chunking sidebands that are positioned just below ± 5000 Hz ($= 1/\tau = 1/(200\text{e-6})$ Hz) is reduced when only the real data points are used (compare the insets in Fig. S2.2b and d); their intensity is at $\sim 3\%$ of the main SHARPER peak. The resonance frequency of the sidebands is slightly lower than 5000 Hz due the additional delays of the order of nanoseconds, which are inserted into the spin-echoes to allow the on/off switching of the receiver. The low level artefacts in the $\pm (2000\text{-}3000)$ Hz regions cannot be fully explained; their broader components can be removed by applying a backward linear prediction (see Fig. S2.2e). Overall, the SHARPER spectra of **1** are very clean; the intensity of the described minor signals is low and inconsequential for the quantification of the SHARPER signal, which sits in the clean central region of the spectrum.

Fig. S2.2. 400 MHz ^1H SHARPER spectra of **1** (see also Fig. 1 of the main paper) produced by processing of the time domain data shown in Fig. S2.1. In (a) and (b), both the real and imaginary data points were used and a line broadening of 0.11 Hz (a matched filter) applied; In (c), (d) and (e) only the real points were used. The insets in (a) and (c) show vertical expansions of the SHARPER signal over ± 1 and ± 40 Hz, respectively. (b) and (d) show a 1024-fold vertical expansion of (a) and (c). Here the insets show 8-fold vertical expansions of the areas around the first sidebands. (e) The same as (d) but with a linear backward prediction of the first 100 points applied to real time domain points only. For comments see the text.

REVIEWERS' COMMENTS

Reviewer #1 (Remarks to the Author):

The authors have answered satisfactorily to all my questions. Congratulations on the excellent paper!